# PISA data clusters reveal student and school inequality that affects results

**Magnus Neuman** *

Integrated Science Lab, Department of Physics, Umeå University, Umeå, Sweden

* magnus.neuman@gmail.com

## Abstract

The data from the PISA survey show that student performance correlates with socio-economic background, that private schools have higher results and more privileged students, and that this varies between countries. We explore this further and analyze the PISA data using methods from network theory and find clusters of countries whose students have similar performance and socio-economic background. Interestingly, we find a cluster of countries, including China, Spain and Portugal, characterized by less privileged students performing well. When considering private schools only, some countries, such as Portugal and Brazil, are in a cluster with mostly wealthy countries characterized by privileged students. Swedish grades are compared to PISA results, and we see that the higher grades in private schools are in line with the PISA results, suggesting that there is no grade inflation in this case, but differences in socio-economic background suggest that this is due to school segregation.

## Introduction

PISA (Programme for International Student Assessment) [1], organized by the OECD, is a 3-year recurring knowledge test used to assess the performance in science, mathematics and reading of 15-year-olds. PISA draws considerable attention worldwide and affects educational policies [2–6]. The 2015 PISA survey, which we focus on here, included 519334 students in 73 countries. The corresponding data set with student responses to the PISA questionnaire also includes a set of items to assess the socio-economic background of the students [7, 8]. This allows for studying the connection between socio-economic background and student performance.

There has been some criticism against PISA, for example regarding the effect of cultural biases in student responses, the lack of adaptations to local curricula and language, and the deceiving simplicity in ranking countries [9, 10], and it has been argued that this criticism is largely ignored [10]. But it has also been shown that the PISA results are indicative of the educational outcome of students [11].

While some studies focus on for example gender differences [12, 13], we here focus on the connection between the socio-economic background of the students and their performance in PISA, since there is growing evidence that this is an important determinant for student

**Data Availability Statement:** Data files are available from https://www.oecd.org/pisa/data/2015database/.

**Funding:** The computations were enabled by resources provided by the Swedish National Infrastructure for Computing (SNIC) at HPC2N

partially funded by the Swedish Research Council through grant agreement no. 2018-05973. The funders had no role in study design, data collection and analysis, decision to publish, or preparation of the manuscript.

**Competing interests:** The authors have declared that no competing interests exist.

performance [14–17], while at the same time the educational system in most cases is designed to compensate for the variations in the students' socio-economic background [18–20]. We therefore explore any possible connections between performance and socio-economic status.

Some alternative analyses to reveal patterns in the PISA data have been done, focusing on the raw data in PISA—the student responses to items (questions)—to study for example longitudinal trends based on item characteristics [21] or country-specific differences using cluster analysis [12, 22]. We believe that the data can be further explored and any possible hidden patterns be discovered using modern methods, mainly from the field of network theory that has developed tremendously over the last decades [23].

In this work we therefore apply tools from this field, previously applied mostly in theoretical ecology on species distribution data [24–28], to find patterns in the PISA data that can tell us something about the structure in performance and socio-economic background of the students and further nuance the PISA results.

We also study the performance of students at private and public schools respectively, since this aspect of school organization also has been shown to affect student performance [29, 30], and grade inflation has been reported at private schools [31–33], in this way reinforcing social inequality and undermining the compensatory effect of the educational system. We compare PISA data to grading in Swedish schools, where we have access to grading data in these two sectors respectively, to investigate possible grading inflation in Sweden.

## Results

PISA consists of a set of items, or questions, that the student should answer in order to assess the student's knowledge in different areas, such as science and mathematics. These items are ranked according to their overall difficulty $d$ which is derived from student responses worldwide [8]. Here we normalize $d$ such that the normalized difficulty $\bar{d}_i$ of item $i$ is $\bar{d}_i = (d_i - \min_j(d_j))/(\max_j(d_j) - \min_j(d_j))$, where $j \in [0, N]$ if $N$ is the number of items. This means that $\bar{d} \in [0, 1]$, which is necessary for further calculations if we want to sum difficulty-weighted responses. The student response $r$ is here coded as either 1 (pass) or 0 (fail). We can then assign a number $P_i$ to a student $i$, giving the student's number of points in science and mathematics, by calculating

$$P_i = \sum_{j \in \{sci, math\}} \bar{d}_j r_{i,j}. \tag{1}$$

This measure then gives a number in a transparent way for each student's achievement in science and mathematics, weighted by the item difficulty.

PISA also includes an index called ESCS (Economic, Social and Cultural Status) derived from the student responses about home possessions and the parents' education and occupation [8]. This index quantifies the students' economic, social and cultural background. We are here interested in the relation between the student's socio-economic background and performance in mathematics and science. It is therefore interesting to study how the students are distributed over P and ESCS, and how this varies between countries and between public and private schools.

Fig 1(a) shows the distribution of all participating students (without student weights and resampling) over points and ESCS. We see that the distribution is not uniform, but skewed so that students with higher points tend to have high ESCS. Highly performing students thus tend to be more socio-economically privileged. Using the student weights [8] to resample the data (details in Methods) we can calculate the correlation (Spearman's $\rho$) between Points and ESCS to be $\rho = 0.2836 \pm 0.0002$, 95% CI. If we consider public and private schools separately

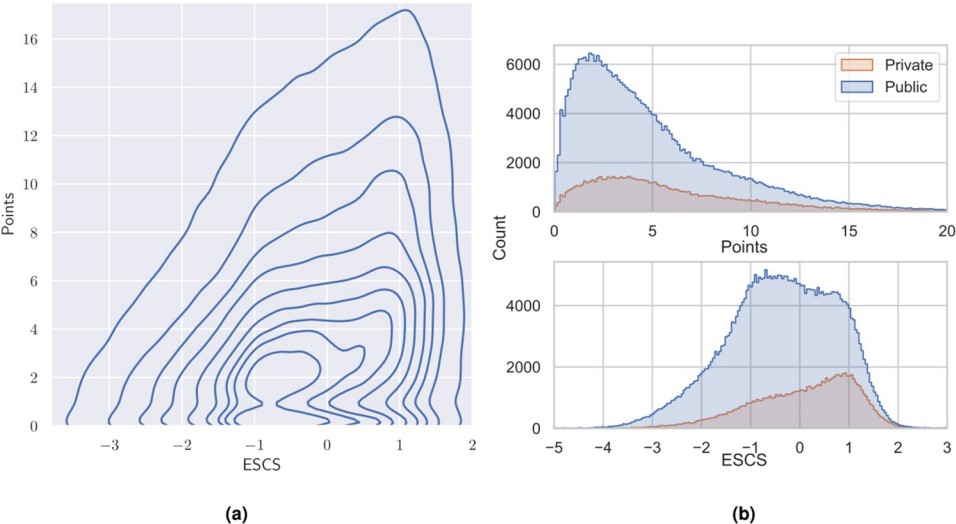

**Fig 1.** The distribution (number of students, as isolines) of student results (Points) over economic, social and cultural status (ESCS) is skewed so that highly performing students tend to be more privileged (a). Fig 1(b) shows that private schools tend to have somewhat better results than public schools (top), and that students at private schools are more privileged (bottom). The 57464 students with zero points (visible in (a)) have been omitted in the top figure to better display the distribution of the other students.

this correlation is $\rho = 0.2777\pm0.0002$ for public schools and $\rho = 0.2246\pm0.0005$ for private schools. The tendency that highly performing students are also socio-economically privileged is thus stronger in public schools than in private schools, on a world-wide level. Another aspect of the differences between public and private schools is shown in Fig 1(b), where we see that the average number of points in public schools is lower than in private schools ($4.366\pm0.001$, 95% CI vs. $5.290\pm0.002$), and that the mean ESCS in public schools is lower than in private schools ($-0.3899\pm0.0003$ vs. $0.0516\pm0.0005$).

As a further step in understanding the differences between countries Fig 2 shows the correlation (Spearman's $\rho$) between points P and ESCS for the participating countries. This ranges from $\rho = 0.313\pm0.002$, 95% CI, for Peru, to $\rho = 0.052\pm0.002$ for Algeria. This shows that there are variations between countries, and that the tendency that highly performing students are also socio-economically privileged is not equally strong in all countries.

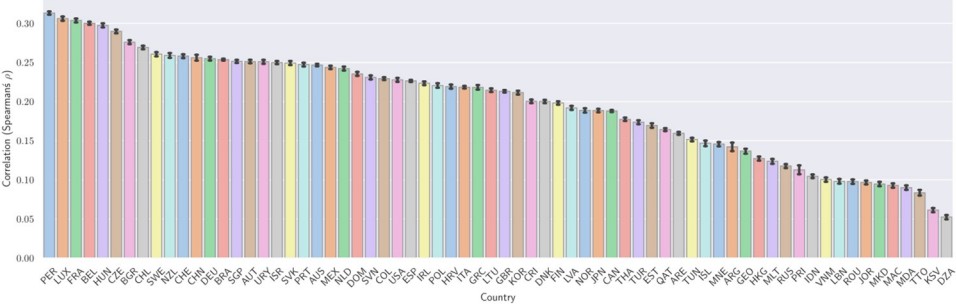

**Fig 2. The correlation (Spearman's $\rho$) between student points in mathematics and science, and student ESCS for the participating countries (ISO 3166-1 alpha-3 codes).** The error bars indicate a 95% confidence interval obtained by bootstrapping the data using the student weights (details in Methods). It is evident that privileged students (high ESCS) tend to perform better in PISA in many countries, but this tendency is weaker in countries to the right in the figure.

This type of data where different groups (here countries) are distributed differently in a possibly multidimensional space can be difficult to interpret based on simple correlations, and the large-scale patterns in the data can be difficult to find. A growing body of research is therefore developing, mainly in the field of theoretical ecology [24–28], to employ methods from network theory to find these patterns by, for example, representing species distributions in geographic space [24–26] or in a space of climatic variables [27] as networks in order to study their structure. We here apply this framework to the PISA data to identify clusters of countries that are connected together due to their students performing similarly in PISA while at the same time having a similar socio-economic background, meaning that their students are distributed similarly in the ESCS-Points space.

To start with we discretize the space of socio-economic background and performance in PISA, i.e. the ESCS-P space, by selecting the number of quantiles to divide each dimension in. This is done by computing the Jensen-Shannon divergence of the distributions of the participating countries in this space, in order to see when it is no longer meaningful to continue to divide the space. Details on this are provided in Methods, where we show that this space should be divided in nine quantiles in each dimension. Each bin in this distretized space is a node in a bipartite network where these space nodes are one category of nodes and the countries are the other category. A country node is connected to these space nodes based on how the country's students are distributed in the ESCS-P space. If the fraction of students in a space node (ESCS-P bin) for a country exceeds the fraction of students in this bin averaged over all countries, the link weight between this country and the corresponding space node is the difference between these fractions. Details on this are given in Methods, and Fig 3 shows an overview of this whole procedure.

Once the network is constructed we employ the Infomap algorithm [34] to find sets of highly interconnected nodes, called modules in network science but commonly referred to as

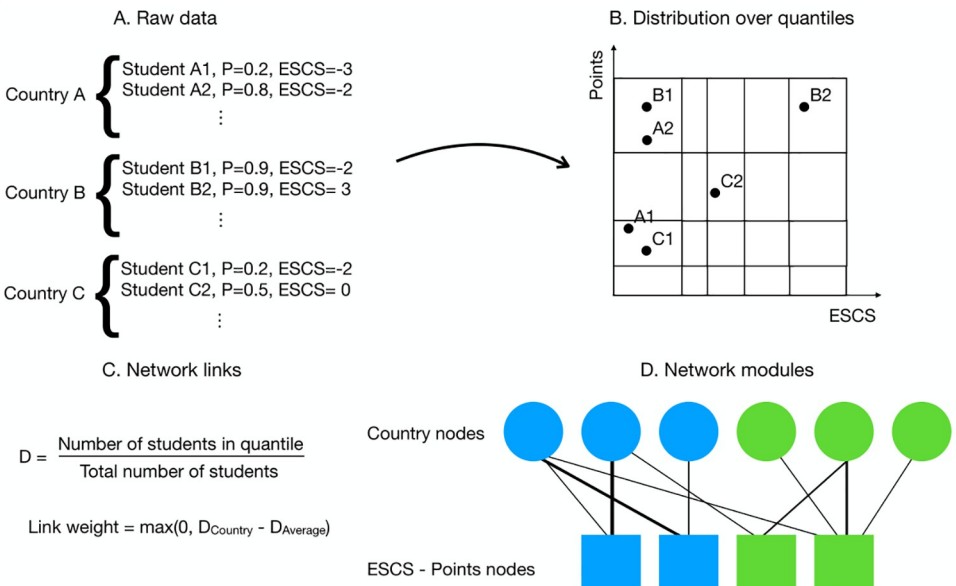

**Fig 3. An illustration of the method used here to find large scale patterns in the PISA data using network theory.** The PISA data (A) are mapped onto a two dimensional space divided in quantiles (B) where each quantile corresponds to a node in a bipartite network with links weights calculated from the data (C). A community detection algorithm (Infomap) is used to find highly interconnected nodes forming clusters (modules) in this network (D). These clusters contain countries with similar patterns in the PISA data.

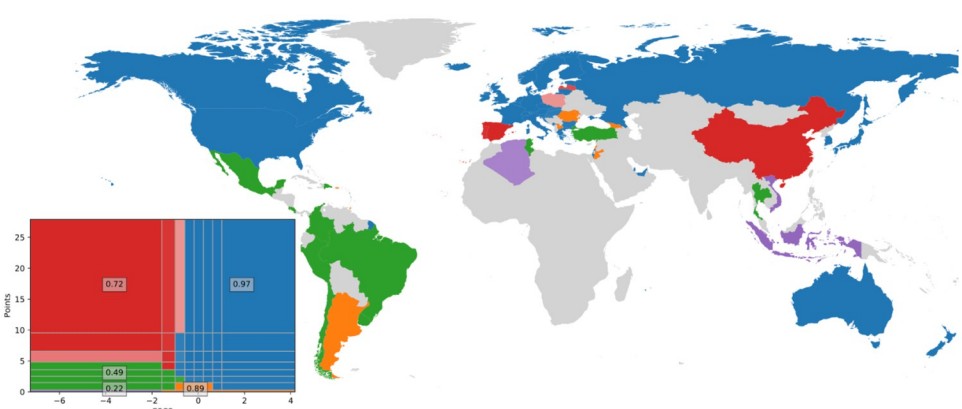

**Fig 4. The clusters resulting from the network representation of the PISA data.** The blue cluster is characterized by privileged students over the whole scale of performance. The green and purple clusters have less privileged students performing poorly. The red cluster has under-privileged students that perform well in PISA. The bootstrap support for the clusters is indicated by the numbers in the inset, and the node significance by the transparency of the colors.

network clusters. Infomap is widely used and has been shown to be highly reliable for clustering networks [26, 35, 36].

Fig 4 shows the resulting clusters of countries and ESCS-Points nodes. The blue cluster is characterized by privileged students on the whole spectrum of performance, and includes many of the world's wealthy countries. The orange cluster has the same range of privileged students but with zero or few points. The green and purple clusters are characterized by less privileged students that perform poorly, with the green having somewhat better performance than the purple. Interestingly, the red cluster is characterized by less privileged students that perform well in PISA. This red cluster includes both Spain and Portugal, China, Latvia and Poland, but the significance analysis (details in Methods) however shows that Poland is less significantly clustered in this red cluster. The significance analysis also shows that both the blue and the red together with the orange cluster have a high support (0.97, 0.72 and 0.89) in the bootstrapped networks (inset in Fig 4).

These clusters of countries can contribute to our understanding of inter-country differences and can nuance the PISA overall ranking [37]. We can see, for example, if the top ranking countries tend to be clustered together and if they share some common feature. We see however that this is not the case. Some top ranking countries like China, including Hong Kong, are in the red cluster, while most of the others are in the blue cluster. In a top ranking country like China the underprivileged students perform well, which the OECD point out in their report [37], stating that they have greater equity than other countries. They also mention Canada in this context but the results shown here indicate that Canada is better characterized by privileged students, together with many other top ranking countries. The OECD does not mention the interesting features of the countries on the Iberian peninsula, Latvia and Poland that they share with China, in that they have high-performing under-privileged students. Countries like Canada and Estonia are ranked high, while many other countries in the same blue cluster are not, such as Sweden and The United States.

If we consider data from all schools, public schools and private schools separately, we obtain three different networks that can be clustered and compared using an alluvial diagram, as show in Fig 5. We see that the clustering of students at private schools is fragmented, resulting in many clusters with low bootstrap support. This is due to the scarce data with relatively few students in private schools worldwide. We can however see some interesting features that have

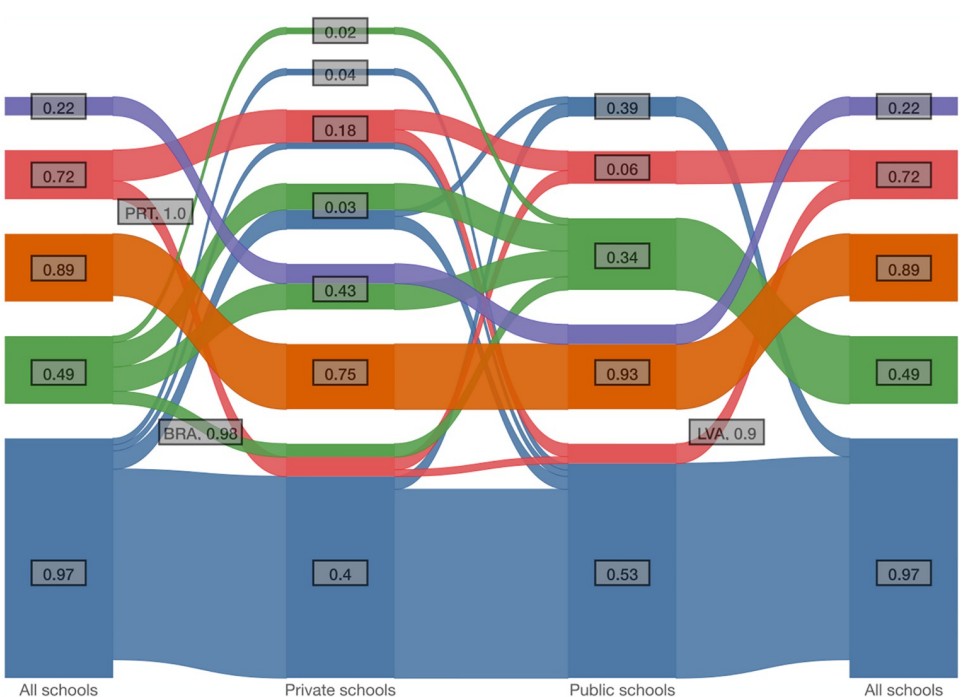

**Fig 5. An alluvial diagram showing how the clustering of countries changes when considering all schools, private schools or public schools, and the bootstrap support of the clusters.** The clustering of countries using data only from private schools is fragmented and some of the clusters have low bootstrap support. We can however see, for example, that Portugal and Brazil both change cluster to the blue cluster, which is characterized by privileged students over the whole scale of performance, when considering private schools only.

support in the bootstrapped data, such as Portugal and Brazil both being clustered in the blue cluster, with privileged students over the whole scale of performance, when considering students from private schools only. Also, Latvia is clustered in the blue cluster when considering only public schools. The other countries in the red cluster remain in this cluster both for public and private schools.

To further study the differences between public and private schools we calculate the median value of points P in each interval of ESCS ranging from -8 to 4 in steps of one. We use bootstrapped data sets to obtain a confidence interval for this median value. Fig 6 shows these data for all countries (a), China (b), Portugal (c) and Brazil (d). In all cases the private schools do not include the tail of socio-economically weak students, but this is most apparent in Brazil. Fig 6 shows that both Portugal and Brazil have significantly higher results in private schools, which was indicated by them switching clusters in the cluster analysis. China has only small differences between public and private schools, but a clear tendency for high-performing students to be also socio-economically privileged. This aspect of China's results goes against OECD's statement of greater equity in China [37]. Portugal and China are clustered together in the red cluster, with high-performing less privileged students, and this is also seen here where students with low ESCS have relatively high points, but for Portugal this is due to the higher performance of these students in public schools. The data for Spain show that students at public and private schools perform similarly.

The world-wide data (Fig 6(a)) shows interesting differences between public and private schools. For example, less privileged students achieve better results in public schools than they do in private schools. At the same time more privileged students score higher in private schools than they do in public schools.

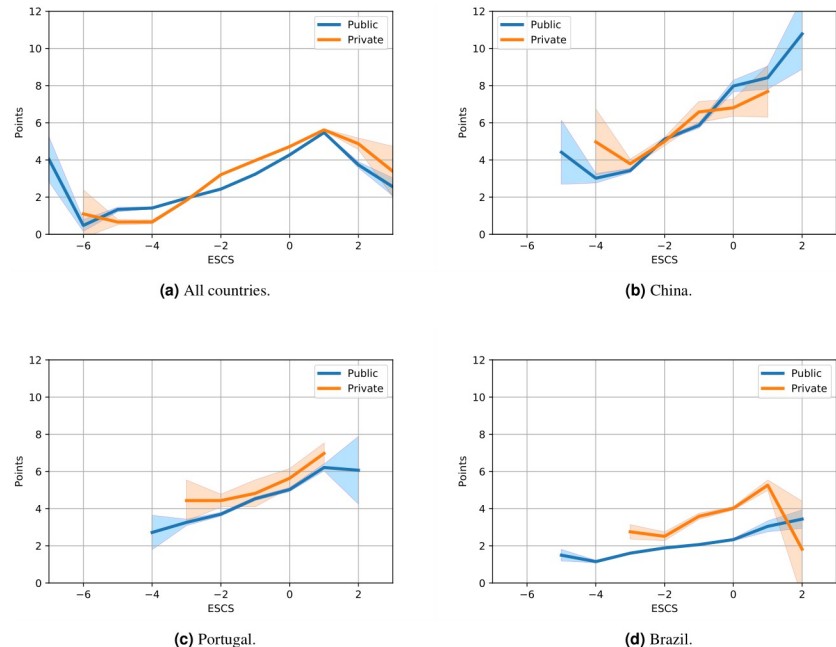

**Fig 6. The bootstrapped median values and confidence intervals for the number of points in different intervals of ESCS.** The network analysis revealed significant differences between public and private schools in Brazil and Portugal, which here shows as higher results (Points) for private schools. Both Portugal and China have relatively high results for less privileged students (low ESCS), which is why they are clustered together when considering all schools.

## School inequality and grade inflation in Sweden

Sweden has seen a highly debated rapid change in educational policy towards market competition in the school system [38–41]. An increase of the inter-school variations in grading of students has been observed [42], and others report that the right to freely choose your school has led to an increase in school segregation [43]. Students at private schools in Sweden are clearly more privileged (ESCS = 0.575±0.005, 95% CI) than students at public schools (ESCS = 0.283 ±0.002), and they also perform better in PISA (Points = 5.79±0.02 vs. Points = 5.26±0.01). We can compare the points in PISA to the corresponding grades in Swedish schools [44], as shown in Table 1 where we include all points, and points in mathematics and science separately. We see that the grading is in line with the PISA results, and these data thus suggest that there is no grade inflation at private schools, contrary to what has been suggested previously [31–33]. In fact, based on the PISA results even higher grades in mathematics can be justified at private schools.

**Table 1. PISA points and the average grades in mathematics, physics and chemistry of students in private and public schools in Sweden.** Students in private schools have better results in PISA and also higher grades than students in public schools. These data thus suggest that there is no grade inflation at private schools in Sweden.

|  | Public Schools | Private Shools | Difference (%) |
|---|---|---|---|
| Points | 5.26±0.01 | 5.79±0.02 | 10.1 |
| Points, Mathematics | 1.450±0.006 | 1.62±0.01 | 11.7 |
| Points, Science | 3.814±0.006 | 4.17±0.01 | 9.3 |
| Grade, Mathematics | 12.0 | 13.3 | 10.8 |
| Grade, Physics | 12.7 | 14.0 | 10.2 |
| Grade, Chemistry | 12.8 | 14.0 | 9.4 |

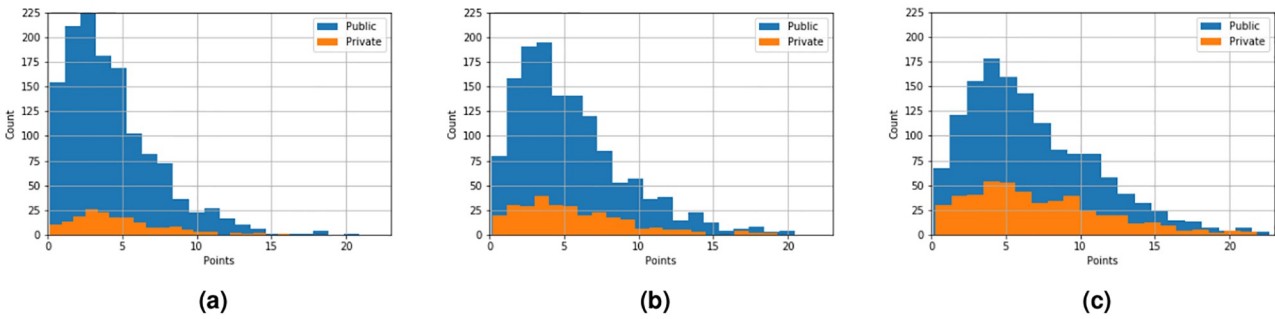

**Fig 7.** Histograms of PISA points divided in three quantiles (low-high in (a)-(c)) of public school ESCS in Sweden show that public schools have a large part of their students in the lowest quantile (a), while the opposite is true for private schools (c). This can explain part of the higher PISA results and grades in private schools in Sweden.

However, by considering how the students are distributed over ESCS in public and private schools respectively, as shown in Fig 7, we see that the public schools have a large part of their students in the lower ESCS quantile, while the opposite is true for private schools. This can explain why private schools have the higher PISA scores (and also grades) that we have seen, since the less privileged students do not attend private schools to the same extent that they attend public schools. This suggests that there is a school segregation in Sweden, where the less privileged students predominantly attend public schools, and that this is responsible for the differences in PISA results and also grades between public and private schools.

## Discussion

We have seen that students from a privileged socio-economic background tend to perform better in PISA and that this tendency varies between countries and private and public schools. These tendencies are known from previous studies, but the network based approach that we propose here sheds new light on this and shows, for example, that some countries go against this general trend since their less privileged students have relatively high results. This group includes China, Spain and Portugal, and further analysis showed that privileged students tend to perform better than less privileged also in these countries, but that less privileged students achieve higher results compared to the world-wide average. In Portugal however, students in private schools perform significantly better than students in public schools. This is not the case in China and Spain. Some countries, such as Brazil, show large differences between public and private schools.

This type of results can nuance the PISA overall ranking and inform policy makers, for example about which student groups to target in order to invest resources in the most efficient way, if the goal is to improve the PISA results in a country. For example, Portugal's public schools perform well but their private schools are significantly better, which should raise questions about the causes of this. Also, Poland and Latvia are clustered weakly in the cluster characterized by socio-economically weak students performing well, and one could ask why this is the case and what can be done if one should desire to be more connected to this cluster.

This way of analyzing difficulty-weighted item responses opens up new ways of doing research with the PISA data. A possible next step in this line of research is to study time resolved (longitudinal) data gathered from several rounds of PISA to see how trends develop over time. It would be interesting to see if there is a detectable signal over time of increasing school and student differences, i.e. segregation, and how this affects the overall results country-wise and world-wide.

A wider contribution of this paper is to advance the understanding of network theory as a tool for data analysis, by contributing with another application of these methods.

The comparison between PISA results in public and private schools in Sweden and their respective grades, showed that the higher grades in private schools are justified given the PISA results. This is evidence that there is no grade inflation in private schools in Sweden since the PISA tests are marked centrally in a standardized manner, thus excluding the role of the school and the teacher which can otherwise affect the results in for example nation-wide but locally marked tests. We however also saw that these differences can be explained by socio-economically weak students predominantly attending public schools.

## Methods

### Data

The PISA 2015 data have been obtained from the OECD website [45] in SPSS format and read using Pandas to do further analysis. Data on item difficulties have been obtained from the PISA 2015 Technical Report [8].

Data on whether a school is private or public is primarily taken from the item "SC013Q01TA", which reads "Is your school a public or a private school?". The exception is Sweden where this information is coded in the field "STRATUM", and values "SWE0001", "SWE0002", "SWE0003", "SWE0004" correspond to public schools. These data are missing for approximately 1/10 of the students worldwide.

### Significance analysis

To estimate summary statistics we perform a bootstrap analysis, as prescribed by the OECD, by resampling the data with replacement to obtain a set of 100 bootstrap samples. Using this set we can calculate for example mean values and confidence intervals in a standard manner. The resampling of students is done with the recommended PISA weights ($W\_FSTUWT$) that compensate for possible biases in the selection process [7]. We have however done extensive testing with and without these weights and the results are in general robust with respect to these weights being used or not.

The significance analysis of modules and module node assignments follows the procedure outlined by Calatayud et al [26]. We measure the distance between two modules $C_i$ and $C_j$ as the Jaccard distance $J(C_i, C_j)$, given by

$$J(C_i, C_j) = 1 - \frac{|C_i \cap C_j|}{|C_i \cup C_j|},$$ (2)

which is one minus the fraction of the number of common nodes and total number of nodes. The significance $\alpha_i^R$ of a module $i$ in a reference partition $R$ is the fraction of partitions that have a module with a smaller Jaccard distance to $i$ than a threshold $\tau$, that is

$$\alpha_i^R = \frac{1}{p-1} \sum_{P \neq R} \Theta \left[ \tau - \min_j J(C_i^R, C_j^P) \right],$$ (3)

where we sum over all $p - 1$ partitions $P$ that are not the reference partition $R$, and $\Theta$ is the Heaviside step function. We have here used $\tau = 0.8$. Due to the bootstrapping procedure with the PISA weights there is no reference partition based on raw or unsampled data. Instead we use as reference partition the bootstrapped (with PISA weights) partition with the shortest average Jaccard distance to all other bootstrapped partitions.

The node significance $\eta_v^R$ of node $v$ in reference partition $R$ is calculated as the fraction of partitions in which $v$ appears in the module that is most similar to $v$'s module in the reference partition. Using the Kronecker delta function $\delta$ this can be written

$$\eta_v^R = \frac{1}{p-1} \sum_{P \neq R} \delta\left(c_v^P, c_v^{RP}\right), \tag{4}$$

where $c_v^P$ is the module index of node $v$ in partition $P$, and $c_v^{RP} = \arg \max_j |C_{c_v^R}^R \cap C_j^P| / |C_{c_v^R}^R \cup C_j^P|$ is the module index of the module in partition $P$ that is most similar to the module of $v$ in partition $R$.

## Dividing the space of socio-economic background and PISA performance

It is important to choose the best possible division of the ESCS-P space to define the nodes in this space. If the division is to coarse grained we loose information, with the extreme case being no division at all and all countries would then be connected to the same node in this space, giving no structure whatsoever. In the other extreme we divide the space in to many quantiles so that only one or very few countries are connected to a node. This also leads to a loss of information.

The best division is somewhere in between these extremes and to find it we calculate the Jensen-Shannon divergence between the countries' distributions in the ESCS-P space, where

$$JSD = H\left(\sum_{i=1}^{N} \frac{p_i}{N}\right) - \sum_{i=1}^{N} \frac{H(p_i)}{N} \tag{5}$$

is the Jensen-Shannon divergence. Here $H$ is the entropy, meaning that

$$H(X) = \sum_i p_X(x_i) \log p_X(x_i), \tag{6}$$

where $X$ is some variable with possible values $x_i$ and $p_X(x_i)$ is the relative frequency of $x_i$, or the probability of $x_i$ if $X$ is a stochastic variable. The unit of entropy is normally called *bits* and it measures the information content in a distribution of $X$, such that a completely uniform distribution has maximum entropy (since it is completely unpredictable if $X$ is a stochastic variable).

The Jensen-Shannon divergence is thus the difference between the entropy of the average distribution of countries, and the average of the entropy of each country's distribution. If this is zero all countries are equally distributed in the ESCS-P space, as is the case if we have only one ESCS-P node. When the number of quantiles is increased *JSD* also increases if there is a difference between how the countries are distributed, but only up to a some point when we gain no or little more information by dividing the space further.

We calculate *JSD* when varying the number of quantiles that we divide the ESCS-P space in and calculate the JSD for each division. Fig 8 shows *JSD* for quantiles in the range 2-16, and we see that there is a steep increase in *JSD* up until some point between 6 and 8 quantiles. To further illustrate this we show in Fig 8(b) the increment of *JSD* as we increase the number of quantiles. We see here that after approximately nine quantiles the increment flattens out, and we therefore choose to use nine quantiles in our analysis.

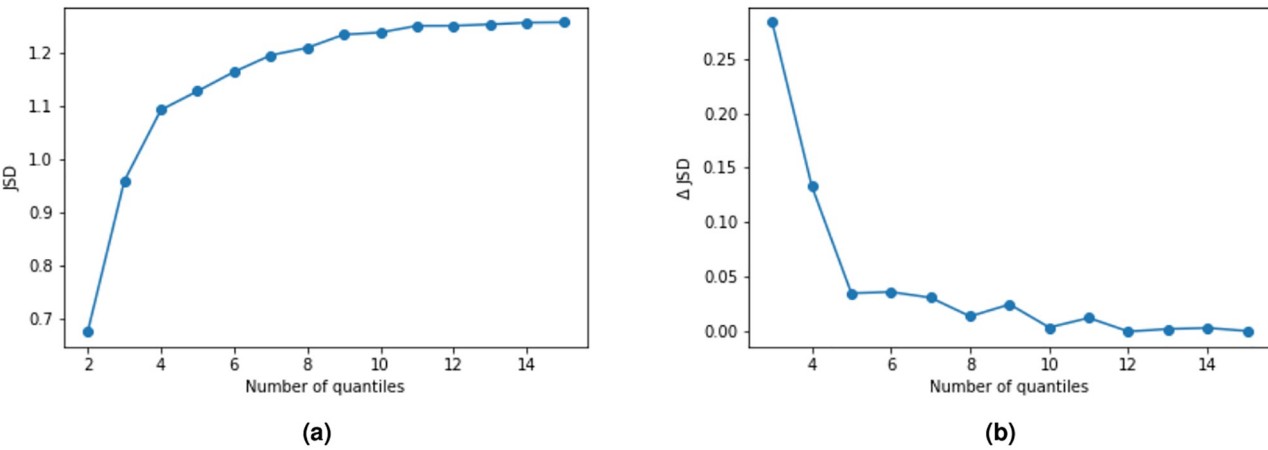

**Fig 8. The Jensen-Shannon divergence (a) and its increments (b) when varying the number of quantiles in the ESCS-P space.** After approximately six quantiles we gain little information by dividing further.

## Building the network of countries and space nodes

Once the space of ESCS and points is divided in quantiles we construct a bipartite network by linking countries to the ESCS-points nodes (space nodes) where the country has a larger fraction of its students than the average of all countries.

The bipartite network is $\mathcal{N} = (\mathcal{C}, \mathcal{S}, \mathcal{E})$, where $\mathcal{C} = \{c_1, c_2, \ldots, c_{N_c}\}$ is the set of $N_c$ country nodes and $\mathcal{S} = \{s_1, s_2, \ldots, s_{N_{E,P}}\}$ is the set of $N_{E,P}$ space nodes. Here $N_{E,P} = q^2$ if $q$ denotes the number of quantiles in each dimension. The set of potential links $\mathcal{E} = \{e_{1,1}, e_{1,2}, \ldots, e_{N_c, N_{E,P}}\}$ thus connects country nodes to space nodes. A link $e_{c',ep'}$ is given by

$$e_{c',ep'} = \max\left(0, \frac{S_{c',ep'}}{S'_c} - \frac{S_{C,ep'}}{S_C}\right) \tag{7}$$

where $S_{c',ep'}$ denotes the number of students in country $c'$ that are in node $ep'$, $S'_c$ the total number of students in country $c'$, $S_{C,ep'}$ the total number of students, in all countries, in node $ep'$, and $S_C$ the total number of students in all countries. This means that there is a link between a country and a space node if the fraction of students of that country in that node is larger than the fraction of students of all countries in that node. A country is thus connected to a space node if the country has a higher proportion of the students in a part of the ESCS-P space than the average, thus capturing the notable features of that country.

## Acknowledgments

N. Eliasson and M. Oskarsson are acknowledged for advice on data retrieval.

## Author Contributions

**Conceptualization:** Magnus Neuman.

**Data curation:** Magnus Neuman.

**Formal analysis:** Magnus Neuman.

**Investigation:** Magnus Neuman.

**Methodology:** Magnus Neuman.

**Software:** Magnus Neuman.

**Validation:** Magnus Neuman.

**Visualization:** Magnus Neuman.

**Writing – original draft:** Magnus Neuman.

**Writing – review & editing:** Magnus Neuman.

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
