## [Decision Letter · Decision Letter 0]

13 Jan 2022

PONE-D-21-35674The PISA data reveal student and school inequality that affects resultsPLOS ONE

Dear Dr. Neuman,

Thank you for submitting your manuscript to PLOS ONE. After careful consideration, we feel that it has merit but does not fully meet PLOS ONE’s publication criteria as it currently stands. Therefore, we invite you to submit a revised version of the manuscript that addresses the points raised during the review process.

ACADEMIC EDITOR: Thank you for submitting your work to Plos one. We have now received two reviews. The reviewers are split in their assessment of the paper. One reviewer, with a background in the technical methods used, suggests that the paper nearly meets Plos criteria for publication. The other reviewer, with significant expertise in the subject area, argues that the manuscript, in its current form, is far from meeting Plos criteria for publication. I am recommending a decision of major revisions. To meet Plos criteria for publication, the author must, at minimum, address the concerns of reviewer one related to the measurement of PISA scores (Reviewer 1’s point 2) and better ground the study in the relevant literature and theory. The statistical approach is novel in this field of study; the author just needs to make clear what new knowledge this approach affords over standard statistical approaches used in previous studies or how this study replicates, in a conceptual, previous results. Note that due to the split reviews and the nature of the reviews, I will seek a third party to review any resubmitted manuscript. Thank you once again for submitting your work to Plos One, and I wish the author well in their research endeavors.

We look forward to receiving your revised manuscript.

Kind regards,

Jacob Freeman

Academic Editor

PLOS ONE

Journal Requirements:

(N. Eliasson and M. Oskarsson are acknowledged for advice on data retrieval. The computations were enabled by resources provided by the Swedish National Infrastructure for Computing (SNIC) at HPC2N partially funded by the Swedish Research Council through grant agreement no. 2018-05973.)

 (The author received no specific funding for this work.)

4. We note that Figure 4 in your submission contain map images which may be copyrighted. All PLOS content is published under the Creative Commons Attribution License (CC BY 4.0), which means that the manuscript, images, and Supporting Information files will be freely available online, and any third party is permitted to access, download, copy, distribute, and use these materials in any way, even commercially, with proper attribution. For these reasons, we cannot publish previously copyrighted maps or satellite images created using proprietary data, such as Google software (Google Maps, Street View, and Earth). For more information, see our copyright guidelines: http://journals.plos.org/plosone/s/licenses-and-copyright.

a. You may seek permission from the original copyright holder of Figure 4 to publish the content specifically under the CC BY 4.0 license.  

Reviewers' comments:

Reviewer's Responses to Questions

**Comments to the Author**

1. Is the manuscript technically sound, and do the data support the conclusions?

Reviewer #1: No

Reviewer #2: Yes

2. Has the statistical analysis been performed appropriately and rigorously? 

Reviewer #1: No

Reviewer #2: Yes

3. Have the authors made all data underlying the findings in their manuscript fully available?

Reviewer #1: Yes

Reviewer #2: Yes

4. Is the manuscript presented in an intelligible fashion and written in standard English?

Reviewer #1: Yes

Reviewer #2: Yes

5. Review Comments to the Author

Reviewer #1: I have the following comments on the paper "The PISA data reveal student and school inequality that affect results."

1. The Methods section should be presented before the Results and Discussion sections.

2. The authors calculated students' "normalized points" Pi by dividing points scored per question by the difficulty of each question. The nominator is the difference between the max difficulty and the min difficulty of each student (page 8). First, however, the max and min difficulties of questions answered vary across students, and the gap between the max and min difficulty also varies across students. Therefore, using the nominator as the gap of max and min difficulty of individual students does not normalize PISA scores across the student distribution. Second, PISA test scores have been weighted to take into account the difficulty of different questions and allow meaningful comparisons across students/cohorts/countries. PISA test score is a better measure of student performance than Pi calculated by the authors. Also, the construction of Pi should be placed in the Methods section.

3. How do the authors define "privileged students" using the ESCS score? What is the cut-off point, and why is that point chosen as a cut-off?

4. Why did the authors use bootstraps (page 7) to calculate summary statistics? The PISA sample is supposed to be representative at the country level. They can use any statistical software to calculate the presented results. From what I read here, the results are the mean of test scores and the correlation between test scores and ESCS for different groups of students.

5. "Significance analysis" (page 7): It's not clear what the purpose of the significance analysis presented in this section is. If I understand correctly, the authors use significance analysis to test for the difference between mean scores across students from different economic backgrounds and/or whether the correlation coefficient between test scores and ESCS is statistically significant. If so, t-test and regressions are methods commonly used in social science for these purposes. If the authors decide to use modules and module node assignments, please cite a peer-reviewed published paper that has used this method to investigate similar issues in education or, in addition, present t-tests and regressions to show that the two approaches are similar.

6. "Building the network of countries and space nodes" (page 9): What is the purpose of using this method? What do the authors mean by "where the country has a larger fraction of its students than the average of all countries"? Do they mean a larger fraction of "privileged students" in the whole student population? If so, the statistic they need to calculate here is the deviation from the mean of the fraction of "privileged students", and that can be calculated using any statistical software.

7. Page 9: What is "Figure ??"

8. The authors need to provide the take-away point from each figure presented in the results section.

9. Figure 1 looks like the authors map PISA's deciles with ESCS scores, but I'm unsure.

10. The authors also conclude that students from privileged backgrounds performing better in Sweden results from school segregation. However, at least in the US, private schools are likely to select students with involved parents, and families with involved parents are likely to have higher test results regardless of ESCS status. Also, to attract students, private schools may provide more educational resources to students and have better teachers than public schools. Can you control for these different effects?

11. Cultures, such as China, where acceptance into top universities is dependent on doing well on exams, means that Chinese students in Public schools are also likely to attend after-school private schools. Is there a way that you can control for students who attend both public schools and private schools?

12. The authors' main conclusions are 1) students with a higher socio-economic status (SES) perform better than students with a lower SES, and 2) the extent of the performance gap varies across countries. The authors should reference more of the literature on higher SES students performing better than low SES students, which is well established in the social science literature. They should also reference the literature on human capital theories that provide explanations for the difference in academic achievement between students from different economic backgrounds.

Reviewer #2: The author analysed PISA data to explore the relationship between students' performance and economic status, searching for differences between public and private schools across countries. He applied network-based tools to construct and cluster a bipartite network formed by countries and portions of a bidimensional space representing students’ performances and economic status. Moreover, the author conducted complementary analysis to further the understanding of network clustering. He found a cluster of countries where less privileged students perform well, being this pattern an exception to the rule.

I think the manuscript is well written, well conducted and state-of-the-art in methodology. The results are robust and the interpretations seem appropriate to the methods and results. My complete inexperience in the topic prevent me to evaluate the relevance of the results, still from a complete layman’s point of view I found the results very interesting. I did not find any point that could be further improved but this two minor suggestions:

I had difficulty following Figure 1a. I think the lines represent isolines, but I am not sure. I would color or number the lines and provide a legend to facilitate understanding.

Please check “Figure ??” in lines 286 and 288

6. PLOS authors have the option to publish the peer review history of their article (what does this mean?). If published, this will include your full peer review and any attached files.

Reviewer #1: No

Reviewer #2: No

---

## [Author Response · Author response to Decision Letter 0]

2 Mar 2022

Rebuttal letter

Response to the Academic Editor

Thank you very much for taking the time to consider my manuscript. It combines seemingly disparate fields and I appreciate the effort to seek out reviewers from different areas of research, and I support your intention to seek out also a third reviewer.

Below follow my comments and corresponding changes. I must mention though that it is my impression that most of the remarks made by reviewer 1 are a result of misunderstandings and/or lack of insight into statistical methods beyond those that are standard in the social sciences. I have however tried to improve the manuscript as requested.

Kind regards,

Magnus Neuman

ACADEMIC EDITOR: […] To meet Plos criteria for publication, the author must, at minimum, address the concerns of reviewer one related to the measurement of PISA scores (Reviewer 1’s point 2) and better ground the study in the relevant literature and theory. The statistical approach is novel in this field of study; the author just needs to make clear what new knowledge this approach affords over standard statistical approaches used in previous studies or how this study replicates, in a conceptual, previous results.

This relates to point 12 of reviewer 1 and is partly addressed in my response to that. I have added another two references [16, 17] that contain results on the relation between educational outcome and socio-economic status. However, previous work offer few nuances beyond stating that academic results improve with increasing socio-economic status (as stated in the introduction), and I believe that the present work can fill that gap.

The main results of the present study (as stated in the abstract, results and discussion) are the identification of a cluster of countries characterized by high performance and less privileged students, together with variations between public and private schools, and the novel approach to studying these data. These results are novel and should be of general interest.

The second sentence of the discussion has been changed in order to further emphasize the contrast between previous work and the present work.

Furthermore, I suggest changing the title to ”PISA data clusters reveal student and school inequality that affects results”, in this way narrowing down the scope of the title and pointing more explicitly in the direction of the novel results.

Response to Reviewers

Reviewer #1

Thank you very much for your valuable comments. I have placed my clarifications and declared the corresponding changes to the manuscript after each of your points below.

I have the following comments on the paper "The PISA data reveal student and school inequality that affect results."

1. The Methods section should be presented before the Results and Discussion sections.

It is commonplace in many journals today, including PLOS ONE, to place the methods section last. This improves readability.

2. The authors calculated students' "normalized points" Pi by dividing points scored per question by the difficulty of each question. The nominator is the difference between the max difficulty and the min difficulty of each student (page 8). First, however, the max and min difficulties of questions answered vary across students, and the gap between the max and min difficulty also varies across students. Therefore, using the nominator as the gap of max and min difficulty of individual students does not normalize PISA scores across the student distribution. Second, PISA test scores have been weighted to take into account the difficulty of different questions and allow meaningful comparisons across students/cohorts/countries. PISA test score is a better measure of student performance than Pi calculated by the authors. Also, the construction of Pi should be placed in the Methods section.

I found two typos in this section and apparently I have also expressed myself unclear. I would like to thank the reviewer for drawing my attention to this. The first remark made by the reviewer is a result of a misunderstanding. The max and min difficulties refer to globally calculated difficulties, including all items/questions for all students, as reported in the OECD technical report. This is therefore a proper normalization of the item difficulties. I have rewritten this sentence and hope that this will avoid similar misunderstandings.

I also found a typo in Eq. 1, where one of the indices was missing, which might have caused confusion. This is now corrected.

In the second remark, the reviewer claims that ”PISA test scores” are a better measure of student performance than the one proposed in the article. Firstly, there is no such thing as ”PISA test scores” on a student level. Secondly, assuming that the reviewer alludes to ”plausible values”, which are reported at student level, there are two arguments against using these: 1. They should not be used to calculate student level results, and 2. This would be significantly less transparent than using the raw data as in the present work. I changed the sentence following Eq. 1 to emphasize this more.

Important parts of the methods are often placed in the Results section. Once again, this is to improve readability.

3. How do the authors define "privileged students" using the ESCS score? What is the cut-off point, and why is that point chosen as a cut-off?

The division of students into groups/clusters is a result of the network representation of the PISA data and the clustering of the network with state-of-the-art methods including Infomap. No claim has been made about the existence of a strict cut-off, and I see no reason to clarify this further. Continuing the analogy with climate regions, what is the cut-off (in terms of eg. precipitation) between tropical and sub-tropical regions?

4. Why did the authors use bootstraps (page 7) to calculate summary statistics? The PISA sample is supposed to be representative at the country level. They can use any statistical software to calculate the presented results. From what I read here, the results are the mean of test scores and the correlation between test scores and ESCS for different groups of students.

The bootstrapping is prescribed by the OECD, as described in the methods section. The first sentence in ”Significance analysis” is now changed in order to make this clearer.

5. "Significance analysis" (page 7): It's not clear what the purpose of the significance analysis presented in this section is. If I understand correctly, the authors use significance analysis to test for the difference between mean scores across students from different economic backgrounds and/or whether the correlation coefficient between test scores and ESCS is statistically significant. If so, t-test and regressions are methods commonly used in social science for these purposes. If the authors decide to use modules and module node assignments, please cite a peer-reviewed published paper that has used this method to investigate similar issues in education or, in addition, present t-tests and regressions to show that the two approaches are similar.

T-tests and regressions are not applicable when working with networks and module node assignments. Instead, significance analyses involving bootstrapping are often used. This is straightforward in fields such as network science, and the reference provided (24, in Phys Rev. E) should suffice. The method we use has the advantage of using the relatively transparent Jaccard index (based on set theory) instead of eg. mutual information (based on entropy).

6. "Building the network of countries and space nodes" (page 9): What is the purpose of using this method? What do the authors mean by "where the country has a larger fraction of its students than the average of all countries"? Do they mean a larger fraction of "privileged students" in the whole student population? If so, the statistic they need to calculate here is the deviation from the mean of the fraction of "privileged students", and that can be calculated using any statistical software.

The purpose of using this method, as stated in the introduction, is that we can benefit from the development of methods in network theory, which have shown useful in eg. theoretical ecology to find large scale patterns in data. The other remarks made by the reviewer are somewhat unclear, but I do believe that the method is sufficiently explained in one figure, the introduction and in the results and methods sections.

7. Page 9: What is "Figure ??”

This is a LaTeX related typo that I now have corrected.

8. The authors need to provide the take-away point from each figure presented in the results section.

I believe that this is already in place.

9. Figure 1 looks like the authors map PISA's deciles with ESCS scores, but I'm unsure.

I do not understand this remark, but I have added a short explanation of Fig. 1a to the figure caption.

10. The authors also conclude that students from privileged backgrounds performing better in Sweden results from school segregation. However, at least in the US, private schools are likely to select students with involved parents, and families with involved parents are likely to have higher test results regardless of ESCS status. Also, to attract students, private schools may provide more educational resources to students and have better teachers than public schools. Can you control for these different effects?

This is a very interesting topic, but unfortunately, I believe the present dataset does not contain enough information to answer these questions.

11. Cultures, such as China, where acceptance into top universities is dependent on doing well on exams, means that Chinese students in Public schools are also likely to attend after-school private schools. Is there a way that you can control for students who attend both public schools and private schools?

I believe this information is not included in the present dataset.

12. The authors' main conclusions are 1) students with a higher socio-economic status (SES) perform better than students with a lower SES, and 2) the extent of the performance gap varies across countries. The authors should reference more of the literature on higher SES students performing better than low SES students, which is well established in the social science literature. They should also reference the literature on human capital theories that provide explanations for the difference in academic achievement between students from different economic backgrounds.

I already reference previous research on this issue, but I have now added further references to previous research [16, 17]. The main conclusion is however the identification of a cluster of countries characterized by high performance and less privileged students, together with variations between public and private schools, and the novel approach to studying these data. This is clearly stated at several locations in the manuscript, including the abstract, but to further emphasize this I suggest changing the title to ”PISA data clusters reveal student and school inequality that affects results”.

Being a multidisciplinary study, the list of referenced literature cannot be exhaustive from the point of view of every related field. One example of this is that I do not relate the findings to human capital theories, and neither do most of the other papers that deal with connections between educational outcome and socio-economical status.

Reviewer #2:

Thank you very much for the time invested and for the valuable comments.

The author analysed PISA data to explore the relationship between students' performance and economic status, searching for differences between public and private schools across countries. He applied network-based tools to construct and cluster a bipartite network formed by countries and portions of a bidimensional space representing students’ performances and economic status. Moreover, the author conducted complementary analysis to further the understanding of network clustering. He found a cluster of countries where less privileged students perform well, being this pattern an exception to the rule.

I think the manuscript is well written, well conducted and state-of-the-art in methodology. The results are robust and the interpretations seem appropriate to the methods and results. My complete inexperience in the topic prevent me to evaluate the relevance of the results, still from a complete layman’s point of view I found the results very interesting. I did not find any point that could be further improved but this two minor suggestions:

I would like to thank the reviewer for these encouraging words. The reviewer gives an accurate summary of the work despite being, as he/she claims, a layman.

I had difficulty following Figure 1a. I think the lines represent isolines, but I am not sure. I would color or number the lines and provide a legend to facilitate understanding.

This is correct and I have clarified this in the figure caption. I however believe that colors and/or numbers would clutter the figure while not contributing significantly to the story, and therefore avoid to include them.

Please check “Figure ??” in lines 286 and 288

This is a LaTeX related typo that I now have corrected.

---

## [Editor Report · Decision Letter 1]

1 Apr 2022

PISA data clusters reveal student and school inequality that affects results

PONE-D-21-35674R1

Dear Dr. Neuman,

We’re pleased to inform you that your manuscript has been judged scientifically suitable for publication and will be formally accepted for publication once it meets all outstanding technical requirements.

Kind regards,

Jacob Freeman

Academic Editor

PLOS ONE

Additional Editor Comments (optional): Thank you for making excellent revisions and for clarifying concepts. The paper now meets Plos One criteria for publication. Best of luck on your future research.
---

## [Editor Report · Acceptance letter]

14 Apr 2022

PONE-D-21-35674R1 

PISA data clusters reveal student and school inequality that affects results 

Dear Dr. Neuman:

I'm pleased to inform you that your manuscript has been deemed suitable for publication in PLOS ONE. Congratulations! Your manuscript is now with our production department. 

Kind regards, 

on behalf of

Dr. Jacob Freeman 

Academic Editor

PLOS ONE